# The Use of Unmanned Aerial Vehicles for Dynamic Site Layout Planning in Large-Scale Construction Projects

**Ahmed W. A. Hammad [1], Bruno B. F. da Costa [2], Carlos A. P. Soares [3] and Assed N. Haddad [4],***

1   UNSW Built Environment, University of New South Wales, Sydney 2052, Australia; a.hammad@unsw.edu.au
2   Instituto Politécnico, Universidade Federal do Rio de Janeiro, Macaé 27930-560, Brazil; bruno.barzellay@macae.ufrj.br
3   Departamento de Engenharia Civil, Universidade Federal Fluminense, Niterói 24210-240, Brazil; capsoares@id.uff.br
4   Programa de Engenharia Ambiental, Universidade Federal do Rio de Janeiro, Rio de Janeiro 21941-901, Brazil
*   Correspondence: assed@poli.ufrj.br

**Abstract:** Construction sites are increasingly complex, and their layout have an impact on productivity, safety, and efficiency of construction operations. Dynamic site layout planning (DSLP) considers the adjustment of construction facilities on-site, on an evolving basis, allowing the relocation of temporary facilities according to the stages of the project. The main objective of this study is to develop a framework for integrating unmanned aerial vehicles (UAVs) and their capacity for effective photogrammetry with site layout planning optimisation and Building Information Modelling (BIM) for automating site layout planning in large construction projects. The mathematical model proposed is based on a mixed integer programming (MIP) model, which was employed to validate the framework on a realistic case study provided by an industry partner. Allocation constraints were formulated to ensure the placement of the facilities in feasible regions. Using information from the UAV, several parameters could be considered, including proximity to access ways, distances between the facilities, and suitability of locations. Based on the proposed framework, a layout was developed for each stage of the project, adapting the location of temporary facilities according to current progress on-site. As a result, the use of space was optimised, and internal transport costs were progressively reduced.

**Keywords:** unmanned aerial vehicle; dynamic site layout; construction planning; optimisation; mixed integer programming; construction management

## 1. Introduction

The construction industry contributes to a significant percentage of the total gross domestic product (GDP) of many economies worldwide s [1]. As construction sites become larger and more complex, it is essential to evolve optimisation techniques that increase the efficiency of the operation of the construction industry [2]. An important factor influencing productivity, safety, and efficiency of construction operations, is the site layout adopted during the various stages of construction. Therefore, an optimised location of temporary facilities that support construction operations on-site is essential for effective construction works [3]. Temporary facilities provide an operational supplementary platform for work tasks taking place within the area of construction, and the process of locating these facilities on-site site is denoted as site layout planning (SLP) [4], or construction site layout planning (CSLP). According to [5], in its most basic concept, the SLP process involves planning, designing, and locating all of the necessary facilities to support the activities carried out during the construction period. However, this static planning concept, established in the early stages of the project [6], where facilities are assumed to have a fixed position throughout the entire duration of a project [7–10], does not reflect the current panorama of the construction operations. Complex projects demand a strong understanding of the

work progress to implement a site layout that meets the needs of construction activities. The flow of materials, equipment, and personnel between temporary facilities must be continuously analysed, reducing operating costs through mathematical optimisation [5]. In other words, decision making must be performed dynamically throughout the various stages of the construction process.

The last three decades have been particularly productive for research aimed at site layout planning [11]. It is important to emphasize that the synchronous development of methodologies such as lean construction and the application of the just-in-time (JIT) concept at construction sites highlighted the planning phase as a critical stage to the success of construction projects, and contributed to the understanding that construction plans must be constantly monitored and corrected [12]. Material stocks reduced to a minimum and resources obtained in periods increasingly closer to the date of use made the incorporation of the time factor into the CSLP problem inevitable. Thus, a new concept emerged, namely Dynamic site layout planning (DSLP). DSLP proposes that site layout plans should reflect changes that constantly take place on site, such as arrival time of materials, the fluctuations in resource demand, the locations where supporting facilities are needed at each stage of the project, and the constant adjustment of construction schedules [2,11]. Thus, inactive space can be reallocated throughout the project, as it becomes available, increasing productivity and making the process more realistic [11,13,14]. This requires the evaluation of the changes that are likely to occur in any given stage of construction. These changes are then implemented in the form of separate layouts that are adopted for each stage of the construction process [15–18]. A challenging aspect of space allocation to facilities over a wide region is the difficulty in identifying and precisely analysing the suitability of available spots on-site. Moreover, due to the nature of the construction environment where progress can vary from the initial plans set out, another facet of DSLP that needs to be considered is the mapping and updating of the site plans to match the actual on-site conditions.

Current practice in the industry mostly relies on daily reports and photos taken by engineers to assess the progress of works against the planned schedules [19]. This procedure can be tedious, time-consuming, and is highly prone to various systematic errors, as it depends on the assertiveness of field professionals, who need to move across the job site in search of relevant information [20]. A solution to this would be to automate the process, where reliance becomes based on a programmed system. One technology capable of being adopted for site layout updating is an unmanned aerial vehicle (UAV) system, commonly known as a drone [21]. A single UAV commonly adopted on construction sites is composed of a remotely controlled aircraft equipped with a highly reliable Global Positioning System (GPS) for location referencing in real time, a control station, and several onboard sensors, including object avoidance technology, and image acquisition and transmission systems [22–25]. For a long time, the use of UAVs was restricted to military operations due to their high acquiring and maintenance costs [12,26,27]. However, in recent decades the technology has evolved significantly, thus leading to a boom in UAV commercial applications [28,29]. The reduction in their size and the increase in their autonomy favoured the use of this device in the most diverse disciplines [30–32], making them capable of performing tasks in a faster, safer, and cheaper way [33]. Some of these applications include traffic control [34], search and rescue missions, landslide monitoring [35], delivery of lightweight items to customers [36], fire detection [37], cultural heritage conservation [38], environmental management, disaster monitoring [39,40], and safety inspection [33,41,42].

The AEC (Architecture, Engineering, and Construction) industry has been one of the most attractive and promising markets for UAVs [23,43], and several researches have explored their potential in a wide range of applications, such as: building inspections [44], site mapping and surveying [27,45], bridge inspection [46], progress monitoring, and site planning [47]. However, despite this, few recent works have addressed the development of methodologies that support DSLP practically through use of UAVs. The main objective of this study is to develop a framework for integrating drones and their capacity for effective

photogrammetry with a site layout planning optimisation model and Building Information Modelling (BIM) for automating site layout planning in large-scale projects, thus facilitating the application of the DSLP concept. A case study that showcases the integration of data acquired via UAVs with a mixed integer programming (MIP) model to solve the SLP was conducted to verify the applicability of the proposed approach.

The remainder of this paper is structured as follows: Section 2 presents the conceptual background of the research, based on a review of recent studies on UAV applications in construction engineering. Section 3 describes the steps needed to map appropriate locations for the site layout planning problem using UAVs. Section 4 presents a numerical case to verify the proposed model and the framework proposed. A discussion based on the case study is presented in Section 5. Finally, Section 6 discusses study implications and summarises the findings, along with work limitations and directions for further research.

## 2. Literature Review

### 2.1. UAV Applications in Construction Planning and Monitoring

The adoption of UAVs for non-military services has been gaining growing interest from researchers [12] and is currently considered relevant equipment in the management of construction sites [48]. The flexibility and manoeuvrability that allow UAVs to access hard-to-reach areas and the skill to cover large-scale sites economically and efficiently [49–51], providing visual access through photos or real-time videos, are some of its greatest attractive features [33,52,53]. These characteristics make this technology ideal for mapping and monitoring activities [1], enabling its use at all construction stages [54], from planning, which requires extensive knowledge of the site conditions, location, and surroundings [47], to permanent monitoring, through information related to the presence of construction resources and workflow progress [55]. As a result, site engineers and site planners nowadays have a tool that helps them obtain detailed and up to date information about the logistics and schedule of construction operations [47].

UAVs have already started to change the way infrastructure is designed and operated and currently represent an estimated market of over USD 100 Billion [23]. The prospects are very promising, given the effort being made to integrate this technology with other systems, including Internet of Things (IoT), Artificial Intelligence (AI), Geographic Information System (GIS), BIM, virtual reality, wireless, Bluetooth [13,23,56]. One particular emphasis of UAV deployment in engineering focuses on 3D modelling, through reliance on photogrammetry and algorithms such as Structure from Motion (SfM) [57]. By mapping the camera's positions and orientation, SfM permits the generation of multiple point clouds that represent the object's geometry, converting aerial images into 3D models [12,50,58]. Thus, the use of UAVs has been considered a new technological revolution in the field of photogrammetry since it presents a high flexibility and low cost, and a complexity of operations compared to traditional methods [30,59]. Martinez et al. [60] optimised the location of risk areas on construction sites using 3D models generated by photographs taken via UAVs. Martínez-Carricondo et al. [58] modelled a historic dam in Spain through the acquisition of photogrammetric data carried out by a UAV.

Several studies have focused on the use of UAVs in conjunction with BIM, not only for 3D modelling but also for 4D and 5D simulations. Digital files obtained from stereoscopic images of the UAV, which also contain 3D ground information, can be integrated with data acquired from Building Information Models (BIM) to better allocate each service facility based on ground conditions, position reference, and proximity to other facilities over the construction time. A comparison between as-planned and as-built progress was provided by Alizadehsalehi et al. [61] using UAVs and BIM. Han et al. [62], used a UAV and BIM to develop an automatic model for monitoring construction progress. Vacanas et al. [19] discussed the usability of UAVs for delay and disruption analysis in infrastructure projects. The authors focused on employing UAVs for providing an in-depth view of the time-related disputes occurring between clients and contractors.

### 2.2. GIS (Geographic Information System) and BIM for Site Layout Planning

Typical site layout planning, for a long time, used to be performed based on 2D plans, involving simplified geometry. When the dynamics of the construction work are considered, a schedule is incorporated in the planning phase to generate dynamic site layouts across the various construction stages [63]. Thus, for generating dynamic site plans, it is essential to be able to generate up to date information for each stage of construction so that changes that need to be implemented on the initial layout of the site can be applied [64]. Such information is often obtained manually by site engineers through on-land captured site images, daily reports, etc. Since the approach is based on ideal design information, the separate layouts generated for the construction stages can be different from layouts that are based on a realistic representation of the actual work being conducted. However, obtaining data for constantly assessing the deviations between as-planned and as-built progress, in order to generate more realistic site layout plans, seems like a tedious task for on-site engineers to carry out. It is the aim of the presented framework to offer the means for providing a more effective method of obtaining the necessary information to produce a dynamic site layout, particularly when it comes to delineating feasible locations. Figure 1 presents the phases undertaken to produce a dynamic layout for each stage of the construction process, via the incorporation of on-site photos obtained from UAVs to generate appropriate facility locations.

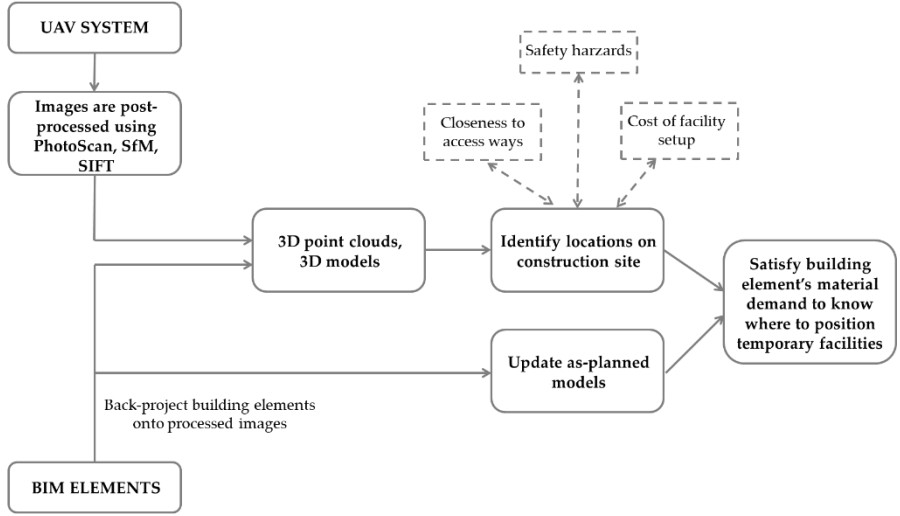

**Figure 1.** Framework linking different modules for SLP planning.

Several systems and techniques have been proposed to facilitate the planning and management of construction site layouts. Research using radio-frequency identification (RFID) and GPS brought interesting contributions to this area of knowledge [65]. However, in recent years, the use of BIM models associated with GIS witnessed a drastic increase in interest amongst scholars in the field, especially in the construction sector [66–69]. According to [70], the use of GIS allows the storage of location data referring to a region or facility that can be integrated with satellite images and digital elevation models (DEM), therefore enabling the analysis of the project's evolution over time [71,72]. Via use of GIS, planners have a tool that helps them in the process of automating the planning of the site layout, modelling their spatial relationships and geometric conditions [73], and contributing considerably to increasing efficiency of site layouts along with reducing construction costs [66].

In addition, the use of BIM for site layout planning has been explored [74]. Incorporating BIM to solve the SLP problem has the advantage of simulating the physical model in a virtual environment [69] through parameterised information, hence, allowing an accurate digital representation of the real object.

Lee and Lee [67] used BIM, GIS and Internet of Things (IoT) to develop a digital twin model for real-time logistics simulation in the construction industry. Pepe et al. [75] proposed the creation of a 3D GIS model of a cultural heritage site combining BIM, GIS, and terrestrial laser scanner. Zhu and Wu [71] developed a common geo-referencing approach for data integration. Liu et al. [68] explored the integration of 4D BIM and GIS during the construction stage, resulting in the term GeoBIM. Khan et al. [69] explored the integration of BIM and GIS for modelling geotechnical properties and safe construction zones based on soil type, which was attemtped in a similar fashion in [73]. Irizarry et al. [76] integrated BIM and GIS to visually monitor the supply chain in construction sites. Finally, [65] used GIS to develop a new methodology for risk assessment on construction sites.

The novelty of the proposed framework in this study for the dynamic SLP can be highlighted as follows. Firstly, the proposed approach combines site layout planning with UAV photogrammetry to identify locations on a construction site for the positioning of temporary facilities; in SLP studies, it is common to assume that the construction site is established as a 2D rectangular space discretised into a grid of candidate locations [53]. It is also common to assume that the locations are a priori declared [77]. Secondly, the proposed framework contributes to the automation of the dynamic SLP in an objective way without emphasis on subjective location decisions, for the dynamic SLP. Specifically, this is the first attempt to link images captured of the site from UAV with mathematical optimisation for the purpose of feasible location identification, through site reconstruction via point cloud. Thirdly, BIM is linked to the images captured via superposition of the as-planned model with the as-built images using well-established techniques [78], thus aiding in the tracking of progress and ensuring that locations mapped as available are up to date. Fourth, the proposed framework allows for the automatic calculation of several optimisation model parameters associated with the locations identified for facility positioning, including the distances between locations and the costs of having locations available for the temporary facilities.

## 3. Materials and Methods

### 3.1. The Use of UAVs in Site Layout Planning

As a preliminary step at the start of each construction stage, aerial photos of the overall construction site are taken through the use of UAVs. These photos are used for dealing with the following: (1) when the construction site is large in size, its terrain can vary from one location to the other. Through rectifying the photographs taken, a digitized terrain model can be formed. Metric photogrammetry can then be employed for the purpose of extracting surveying information. Terrain data can then be used to identify the suitability of a location for a given temporary facility. The photos can also serve the purpose of delineating the construction work zones based on safety requirements; (2) as the project progresses it is expected that changes would occur to the overall structure of the construction site. Photos taken at regular intervals from a UAV allow the decision maker to view the change that is occurring and permit the use of physical measurements from post-processed imagery to determine the suitability of the layout to be adopted. (3) During construction, references for building elements of the constructed structure can be obtained for the geo-referenced images. Identifying such information is critical for understanding how the progress of as-planned vs. as-built designs compares. Changes that are highlighted can easily be incorporated in such designs.

The UAV component, embedded within the framework, is composed of two main modules, namely hardware/software and image processing, each contributing to the generation of the required data of the construction site (Figure 2). The final processing of all data acquired from UAV images and from as-planned designs such as BIM allows for a dynamic site layout to be generated for each stage of the construction process, based on minimising the total material handling costs between the temporary facilities. Figure 2, therefore, demonstrates how information that is generated from UAVs can be easily integrated into the SLP to enhance the accuracy of the location parameters embedded in the SLP mathe-

matical models. In this sense, unlike works that focus specifically on practical applications of UAVs, our study focuses on the development of a framework for linking data obtained from UAVs into the SLP mathematical model, for better representation of the parameters of the model associated with the location decision variable.

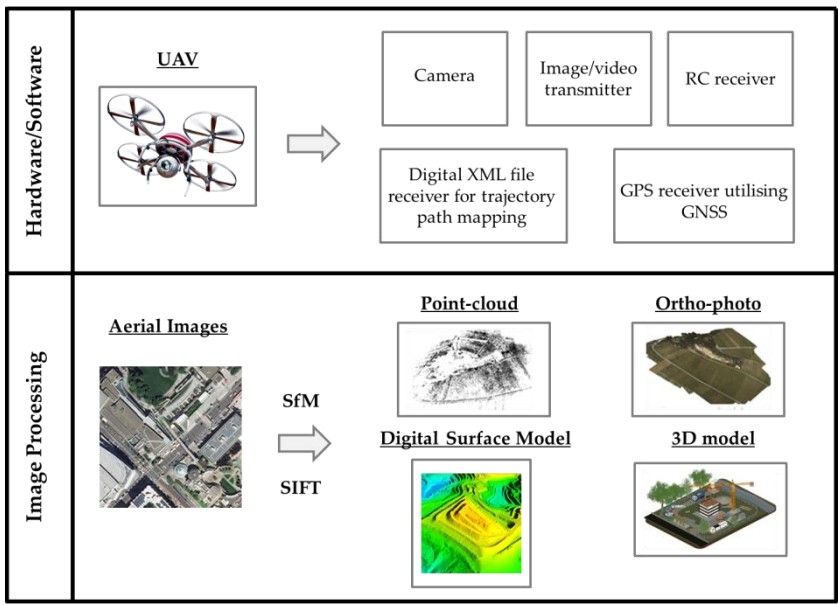

**Figure 2.** Components of the UAV system.

For large-scale projects, especially for infrastructure projects or residential developments, the stretch of works undertaken can be large enough such that a detailed plan of the dynamical changes undertaken is demanded. As progress proceeds, essential facilities required during the construction stage for supporting the work task will need to be positioned in places deemed optimal for ensuring that work disruptions are kept to a minimum. Reducing the total transportation costs involved in moving materials between facilities is the main driver of the optimisation model presented in this paper. Materials are delivered between facilities during the construction stages, and it is through these materials that elements within the structure/building can be constructed. Other facilities such as generators and batch plants provide main services to facilitate the construction process. Many temporary facilities require a certain set of criteria for their potential placement areas to be met before being assigned the position. Examples of these criteria include safety, closeness relationship to a certain topography for ease/restriction of access (security), noise limits, etc. [63].

Three main areas that have received major technical advancements recently form the underlying foundations of the proposed framework for SLP. These are labelled as follows: (1) Imagery from UAVs [79]; (2) Progress derivation detection using BIM [63]; (3) Linking of BIM and UAV for the optimisation of the site layout planning problem [80]. Overall, to ensure that the temporal effects of the construction stages on the development undertaken on a construction site are well accounted for, a work schedule is also incorporated. The assimilation of the principal components of the proposed framework is presented in Figures 1 and 3.

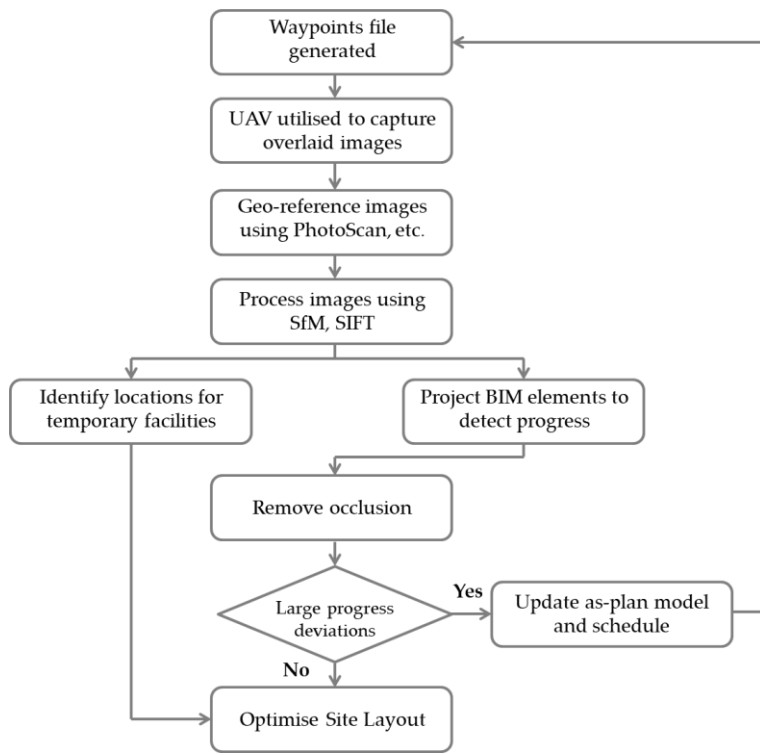

**Figure 3.** Flow chart depicting site layout updating process.

As shown in the framework of Figure 1, data for the purpose of producing a site layout plan over the construction period rely on three main aspects. Firstly, images taken by the UAV system are post-processed to determine the applicability of available locations for locating facilities in. Secondly, back projection of BIM elements is applied onto the geo-referenced images of the UAV system to update as-planned designs and to ensure that they are a representative case of the actual progress. This is achieved by comparing the 3D model generated using the overlayed aerial images taken by the UAV and the Building Information Model of the project involved. The third part relates to the use of information from updated BIM and from ortho-photos and 3D point clouds to generate parameters for the site layout problem.

### 3.2. Generation of Locations for SLP

A key contribution of this study is describing how UAVs can be implemented for the purpose of generating the appropriate locations available for placement of temporary facilities on a construction site, along with the computation of location parameters in the mathematical optimisation model involved. The steps involved in the generation of the dynamic site layout of a large construction site are summarised in the flowchart of Figure 3. The process addresses integrating UAV with the as-planned models such as BIM so that site layout mapping and updating across the different stages of a project is achieved.

Incorporating UAV technology for site layout planning enables the consideration of physical measurements that are directly extracted from images taken at a set interval by a mounted camera. Other aspects accounted for include the consideration of varying site conditions in terms of location availability that can quickly be incorporated in the site layouts of each stage of the project, thus leading to a dynamic generation of the set of available locations for facility positioning throughout the construction project phases. The approach proposed relies on use of UAV as they are regarded as an efficient and cost-effective alternative technology for monitoring the evolving process of construction projects. Assessing the impacts of ongoing work on the location of temporary facilities can be fully examined when regular updates of the occupied areas on the site are available.

Being able to closely monitor changes on the construction site due to applications of the UAV system also offers the ability to update and verify the frequency of travel between facilities on the construction site. This, as explained later in the paper, comprises a vital part of the optimisation model used for SLP.

As an initial step in Figure 3, it is vital to supply the UAV system with a waypoints digital XML file for delineating the flight trajectory path. The path should be programmed such that essential coordinate points in the physical space, including longitude, latitude, and altitude coordinates, are incorporated in the UAV's travel trajectory. A preliminary set of coordinate points can be obtained from initial 2D plans of construction works at the early stages of construction, and from the as-planned 3D BIM of the project to be constructed. Waypoints are defined based on important field views that form the border of the area to be investigated by the UAV. Once acquired, the waypoints may be subject to updates based on a comparative study conducted to assess deviations in the planned *vs.* actual progress of works.

Embedded within the UAV system is a navigation routine that is comprised of a GPS, making use of Global navigation satellite system (GNSS) and an inertial navigation system (INS). Such systems enable the autonomous tracking of the defined waypoints. Apart from its use in navigation, incorporating a GPS receiver within the UAV system also serves the purpose of geo-spatially referencing data acquired from captured images. This is imperative for the construction of ortho-photos from which direct measurements can be made for the optimisation model's parameters, such as distances, areas, and coordinates of the centroids of the locations used to solve the SLP problem; this step is important too for exporting of data to DEM that enable an estimate of the cost parameter in the optimisation model required to prepare a location for hosting a temporary facility. Referencing the data acquired from images captured, therefore, is necessary as it enables the identification of desirable locations for temporary facilities and the production of an updated site layout configuration that is labelled based on accurate coordinates.

Processing of images captured by the mounted camera on the UAV is conducted to produce four types of data visualizations that will be imperative for the generation of several location parameters embedded in the SLP model. For generating 3D point clouds, Scale Invariant Feature Transform (SIFT) is applied to allow for the detection of key feature points within the delineated zone on the construction site. These point clouds are directly geo-referenced data points, generated from densely grouped coordinates. From the point clouds, the DEM and the ortho-photos can be formed. Utilising the ortho-photos and the DEM, locations on the construction site can be analysed based on distance measures between the locations and material demand points, and suitability of terrain for construction of the temporary facility can then be used to generate the cost associated with each location on site. The costs generated relate to the terrain of the location in terms of the setup costs required to be expended for placement of a temporary facility there. This cost is later on embedded in the objective function as a parameter associated with each location identified, thus allowing decision makers to minimise the total monetary cost of setting up the site layout.

For site layout planning, it is important to identify any safety issues and hazards linked with the construction works. These can be pinpointed from the processed images obtained from the UAV technology. Special safety requirements essential to some temporary facility types, for example, the placement of engineers' offices as far away from falling hazards as is permitted, can be accounted for through viewing suitability of locations and their closeness to hazardous areas on ortho-photos. A negative weighting can then be assigned to these locations in the optimisation model parameters generated, to avoid locating facilities in the hazard-deemed regions.

To infer how the progress on site compares with the planned schedule, the 3D point clouds processed from images taken by the UAV system can be superimposed on the Building Information Model. Each building element of a structure undergoing construction can be back-projected onto the processed images for occupancy-based and material-based

appearance modelling. Detection of any mismatch between as-built data, acquired from UAV images, and as-planned models, such as BIM, will entail the updating of the planning schedule and as-planned models to match actual site progress. This will then determine whether the construction site layout should be updated, or whether certain facilities need to remain in particular regions for a given period. Occlusions present in the captured images are dealt with when back-projecting BIM elements against the images, and this can lead to the specific information being modelled and fed into the optimization module for determining whether the progress of a certain element/group of elements requires the repositioning of facilities. Gaps in the information generated from both spectra can therefore be filled when contrasts are made between as-built and as-planned designs.

## 4. Results and Discussion

An industry partner performed the photogrammetry part for the purposes of examining the integration of UAVs with site layout planning, using a DJI Inspire 1 v2.0, whose technical details are given in Table 1. As can be noticed from the technical details of the UAV, a simple drone was used. The framework proposed, thus, does not require a sophisticated system and the basic requirements of a suitable UAV for the framework can be readily available anywhere in the world. An autonomous flight mission was defined using DJI GS Pro for the flight plan and the setting up of flight restrictions.

**Table 1.** UAV technical details.

| Parameter | Value |
|---|---|
| Dimensions | $43.8 \times 45.1 \times 30.1$ cm |
| Weight | 2.93 kg |
| Camera resolution | UHD (4K): $4096 \times 2160$ p24/25 |
| Speed | 22 m/s (max) |
| Field of view | $94°$ |
| Battery capacity | 4500 mAh |
| Wind resistance | 10 m/s |

Once potential locations are identified by the processed images of the UAV, and once the progress of work has been monitored such that an updated work schedule and as-planned building model are produced, the next task is to use the data and parameters generated from the captured UAV images to optimise the layout of facilities. In order to achieve this, a mixed integer programming (MIP) model is formulated and applied to a large civil works project, involving the construction of two major terminals in Kuwait for an airport (Figure 4) (See Appendix A for notation). The UAV system is assumed to be deployed for two stages of the project to which the MIP model is applied to obtain the optimised site layout. The objective function (Equation (1)) minimises the total transport cost between the temporary facilities, as this is one of the most important measures determining the suitability of the constructed layout. Specifically, to compute the transport cost, the frequency of travel between facilities, $F_{ijt}^g$ is multiplied by the distance $D_{mn}$ between locations where facilities are positioned, based on the decision variables $z_{im}z_{jn}$. The cost of preparing location $m$ for accommodating facility $i$ is represented by $c_m$.

$$\text{minimise} \sum_{i,j\in F} \sum_{m,n\in L} \sum_{g\in G} \sum_{t\in T} C_t F_{ijt}^g z_{im} z_{jn} D_{mn} + \sum_{i\in F, m\in L:\tau_{im}=1} c_m z_{im} \tag{1}$$

A number of constraints are also formulated to define the feasible region. In particular, the allocation constraints ensure the placement of all temporary facilities in exactly one location (Equation (2)), so long as the location is suitable for placement of facility $i$ based on $\tau_{im} = 1$.

$$\sum_{m\in L} z_{im} = 1 \forall i \in F : \tau_{im} = 1 \tag{2}$$

The boundary constraints impose the condition of locating the temporary facilities within the delineated location space allocated (Equations (3)–(6)). Specifically, the boundary

of the facility (determined by centroid ($c_i^x$, $c_i^y$) and width, height of facility ($Wf_i, Lf_i$)), must be within the boundary of the location (centroid ($CLX_m, CLY_m$) and width, height ($WL_m, LL_m$)).

$$c_i^x + (0.5Wf_i) \leq (CLX_m + 0.5WL_m)z_{im} + W(1 - z_{im}) \forall i \in F \forall m \in L \tag{3}$$

$$c_i^x - (0.5Wf_i) \geq (CLX_m - 0.5WL_m)z_{im} \forall i \in F \forall m \in L \tag{4}$$

$$c_i^y + (0.5Lf_i) \leq (CLY_m + 0.5LL_m)z_{im} + B(1 - z_{im}) \forall i \in F \forall m \in L \tag{5}$$

$$c_i^y - (0.5Lf_i) \geq (CLY_m - 0.5LL_m)z_{im} \forall i \in F \forall m \in L \tag{6}$$

The overlap constraints prevent facilities from occupying the same space at the same stage (Equations (7)–(9)). This is again based on the boundary of the facility (determined by centroid ($c_i^x$, $c_i^y$) and width, height of facility ($Wf_i, Lf_i$)). The condition in Equation (9) states that either the overlap is prevented in the horizontal direction $\mu_{ij}^x = 1$, or the vertical direction $\mu_{ij}^y = 1$, or both when a facility is placed in location $m$.

$$\left|c_i^x - c_j^x\right| \geq 0.5(Wf_i + Wf_j) \cdot \mu_{ij}^x \forall i, j \in F_T : i \neq j \tag{7}$$

$$\left|c_i^y - c_j^y\right| \geq 0.5(Lf_i + Lf_j) \cdot \mu_{ij}^y \forall i, j \in F_T : i \neq j \tag{8}$$

$$1 + \mu_{ij}^x + \mu_{ij}^y \geq z_{jn} + z_{im} \forall i, j \in F_T : i \neq j \forall m, n \in L : m = n \tag{9}$$

A single site layout is produced for each construction stage considered, hence rendering the layouts dynamic. Using information from the UAV and after updating the as-planned designs, parameters that are derived for the case example include the available locations, their proximity to access ways, distances between the centroids of the locations, the distance between the locations and the structure undergoing construction, and suitability of locations concerning facilities to be allocated. These parameters are updated at each stage of the construction considered. This ensures that the data fed into the optimisation model is an accurate representation of the actual progress occurring on a construction site. The frequency parameter $F_{ijt}^g$, which is an integral parameter in the objective function, is partly produced through distance data extracted from UAV's processed images.

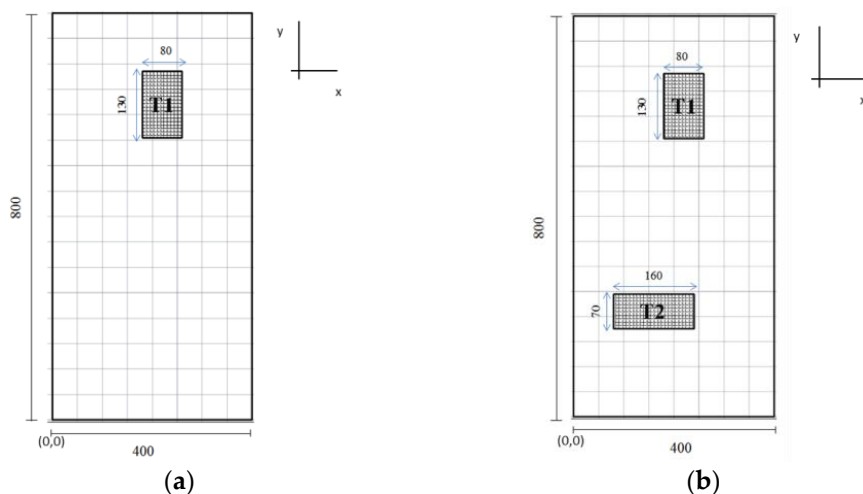

**Figure 4.** Plan view of construction project. (**a**) Stage 1: Construction of Terminal 1; (**b**) Stage 2: Construction of Terminal 2.

Each identified location is assessed by looking at the ortho-photos produced and the DEM so that a cost can be associated with the relating facility construction setup. The cost directly reflects the terrain of the identified location and its closeness to access ways on

the construction site. A mapping between the facilities and the locations then ensues and this is formulated into the MIP model as an additional cost parameter (apart from material handling costs).

Figure 4 shows a plan view of the airport project; in Figure 4a, Stage 1 of the construction process entails the construction of Terminal T1, whereas Stage 2 is defined by the start of construction works for Terminal T2. Global optimum results of the MIP model are presented in Table 2.

**Table 2.** Site layout for two stages of the construction process.

| Facility | Position (Stage 1) | Position (Stage 2) |
|---|---|---|
| Steel yard | (325,523) | (243,321) |
| Storage 1 | (357,530) | (196,281) |
| Storage 2 | (78,678) | (105,221) |
| Generators | (231,596) | (41,121) |
| Formwork yard | (227,632) | (163,200) |
| Offices | (389,450) | (389,450) |

As can be seen from the table, a dynamic SLP is generated, where two separate layouts are produced for each stage of the construction process. At Stage 1, the feasible space on the construction site is defined by all areas, excluding the region occupied by Terminal T1. At Stage 2, the positions occupied by Terminal T1 and T2 are deemed unavailable for location of the temporary facilities. Two frequency parameters are utilized, one for each stage of the construction process, to account for changes identified through data processed from the UAV system. At each stage the distance parameter between locations, $D_{mn}$, is also modified, taking into account the change that results due to the altered site conditions as construction progresses. Initially, the location of T2 was assigned at coordinates (355,163) in the 2D planar space. However, once excavation works commenced it was noted that the soil conditions were varying drastically from the initial geotechnical reports. As a result, this required the updating of the as-planned designs so that T2 was shifted from its original position. Another concern raised during the construction was the alteration of the dimensions of T2 to meet the request of the client. With the incorporation of the framework presented in this paper, all these modifications can be implemented so that as-planned designs match as-built models. Parameters embedded in the MIP model were, thus, a realistic representation of actual site conditions, and the results of Table 1 are suitable for use in SLP.

## 5. Discussion

During the execution of the case study, a number of points were observed. First, there were certain challenges related to the adoption of the method by the working team on the project. This was found to be due to the training needed in terms of drone operation, use of image processing to generate the required parameters, integration of point cloud with as-planned models for dynamic schedule validation, and linking of data extracted from the BIM with the optimisation model. When it came to the drone operation, there were some misconceptions and misunderstandings in terms of the operating duration of the drone. The research team made it clear that if the camera was set to capture an image every 2 s, then, in the span of less than 10 min, the entire construction site could be covered, and the resulting images that are stitched together to produce the point cloud would be of high resolution. The battery capacity of the UAVs adopted should, therefore, be at least 12 min, approximately, in duration, which is satisfied on most of the drones available on the market with similar specifications as Table 1. What was also noticed is that many of the engineers and architects working on the project had limited understanding of drone technology, along with their poor comprehension of the benefits of deploying operational research techniques to conduct an optimised dynamic site layout plan.

Another important challenge that is likely to be experienced on projects implementing the proposed framework is clutter; specifically, obstacles may obscure a clear vision of the camera, and this can block certain aspects of the site from being observed. The data processing step in terms of generating the point cloud model is highly dependent on the amount of pictures that are captured and on the size of the project. For site layout planning problems, since the importance lies in the mapping of the outdoor environment, not much emphasis is placed on creating high resolution models for the building elements themselves, so long as outdoor site conditions can be mapped for pinpointing suitable locations available to position the facilities.

A key point to emphasise is that the method is very easy to implement and does not require the use of a sophisticated UAV type. Most drones available on the market have the minimum technical capacity required to enable such a framework to be implemented. It is also important to note the significance of the validation stage that is performed when the as-built point cloud model is contrasted with the as-planned BIM models; this step enables the verification of the progress of the project, thus allowing for a correct mapping of the stage of the project, and so an SLP problem can be solved that is explicit for that stage.

Despite the benefits generated by the use of UAVs in construction engineering, there are still several technical and managerial challenges that must be overcome for the technology to continue advancing in the sector [54]. Concerns about the safety of using drones in densely populated job sites continue to be identified as the biggest barrier to the spread of these devices, since an operator error can result in serious accidents, with personal and property damage [33,47]. In addition, legal aspects such as property rights and invasion of privacy are also the subject of ethical discussions [33]. Technical issues are also on the agenda of researchers and enthusiasts of this technology. Problems related to battery life, the need for training, the difficulty or impossibility of flying in certain weather conditions, such as strong winds, and the quality of information obtained in these situations are currently the object of study and should be improved so that application of UAVs will gain more use in the construction industry [47,54].

One final note to make is that since the present work seeks an optimal solution to the proposed site layout planning problem through mathematical optimisation, a verified solution is reached once convergence in the solution algorithm is achieved (i.e., the gap between upper bound and lower bound on the solution is zero). It is impossible to cross-validate solutions on a large site as it would require trying all combinations of possible facility–location allocations in reality, which no organisation would agree to due to the costly nature of the process.

## 6. Conclusions

A framework was presented for incorporating a UAV system to aid in the process of site layout mapping and updating on large construction sites. The UAV system captures overlaid images from different positions within the site, as determined from an input waypoints file. These images are processed to produce 3D point clouds, ortho-photos, digital surface models, and, consequently, 3D models of the constructed structures. Data on the locations available for positioning facilities, including their centroids, associated costs, and distance parameters extracted from the 3D point clouds, ortho-photos, and digital surface models, were then embedded in an MIP optimisation model formulated to minimise the total material handling costs and the temporary facility construction setup costs. A single layout is produced for each stage of the construction process, hence rendering the SLP problem a dynamic one. The use of UAVs in the site layout planning problem addressed allows the generation of accurate locations available across the stages of the project, via effective coverage of large-scale sites efficiently and economically. This allows the BIM model to be updated with greater frequency, consistency, and accuracy, reducing the chances of human error in the mapping processes. Applications of the framework were illustrated on a large civil works project.

The limitations of the study are as follows: First, the framework developed, even though it permits dynamic site layout planning to be performed, lacks real-time capacity to capture site progress and make suggestions as to what needs to change in terms of the site layout for the next construction phase. Such capabilities can be permitted if machine learning algorithms are integrated with the UAV system, which was not within the scope of the existing study. Second, the proposed system is not fully automated in that it requires manual input for defining the flight mission, processing the images from the UAV to generate the point cloud, and identifying suitable locations from the BIM model. To address such gaps requires coding of all steps into an automated platform, which is currently being undertaken by the authors.

**Author Contributions:** Conceptualization, A.W.A.H. and A.N.H.; methodology, A.W.A.H.; software, A.W.A.H.; validation, A.W.A.H., A.N.H. and C.A.P.S.; formal analysis, A.N.H. and C.A.P.S.; investigation, B.B.F.d.C.; resources, A.W.A.H. and A.N.H.; data curation, A.W.A.H. and B.B.F.d.C.; writing—original draft preparation, A.W.A.H. and B.B.F.d.C.; writing—review and editing, B.B.F.d.C. and A.N.H.; visualization, A.W.A.H. and B.B.F.d.C.; supervision, A.N.H. and C.A.P.S.; project administration, A.N.H. All authors have read and agreed to the published version of the manuscript.

**Funding:** This research received no external funding.

**Institutional Review Board Statement:** Not applicable.

**Informed Consent Statement:** Not applicable.

**Data Availability Statement:** Data will be available upon reasonable request.

**Acknowledgments:** Assed Haddad wants to acknowledge research grants from CNPq (Conselho Nacional de Desenvolvimento Científico e Tecnológico), Brasilia, DF, Brazil (the Brazilian National Research Council), and Fundação Carlos Chagas Filho de Amparo à Pesquisa do Estado do Rio de Janeiro (FAPERJ), which helped in the development of this work.

**Conflicts of Interest:** The authors declare no conflict of interest.

## Appendix A

**Table A1.** Set notation employed in the MIP model.

| Notation | Description |
|:---:|:---:|
| $F$ | Set of all temporary facilities to be allocated a position on site |
| $L$ | Set of all available locations within which temporary facilities will be allocated |
| $G$ | Stages in project schedule, indexed by $g$ |
| $T$ | On-land transportation equipment, indexed by $t$ |

**Table A2.** Parameter notation employed in the MIP model.

| Notation | Description |
|:---:|:---:|
| $F_{g_{ijt}}$ | Travel frequency of transportation equipment $t$, from facility $i$ to facility $j$, during stage $g$ |
| $C_{rt}$ | Cost of operating transportation equipment $t$ |
| $\tau_{im}$ | Equals one if location $m$ is suitable for locating facility $i$ |
| $W$ | Width of construction site, in the horizontal $x$ direction |
| $B$ | Length of construction site, in the vertical $y$ direction |
| $Wf_i$ | Width of facility $i$ in the $x$ direction |
| $Lf_i$ | Length of facility $i$ in the $y$ direction |
| $D_{mn}$ | Distance between locations $m$ and $n$ |
| $CLX_m$ | $x$-coordinate of centroid of location $m$ |
| $CLY_m$ | $y$-coordinate of centroid of location $m$ |
| $WL_m$ | Width of location $m$ in the horizontal $x$ direction |
| $LL_m$ | Length of location $m$ in the vertical $y$ direction |

**Table A3.** Decision variables employed in the MIP model.

| Notation | Description |
| --- | --- |
| $z_{im} \in \{0,1\}$ | Equals one if facility $i$ is at location $m$ |
| $c_i{}^x \geq 0$ | $x$-coordinate of centroid of facility $i$ |
| $c_i{}^y \geq 0$ | $y$-coordinate of centroid of facility $i$ |
| $\mu_{ij}^x \in \{0,1\}$ | Equals one if facility $i$ and $j$ do not overlap in the horizontal $x$ direction |
| $\mu_{ij}^y \in \{0,1\}$ | Equals one if facility $i$ and $j$ do not overlap in the vertical $y$ direction |

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
