# Peer review of "The Use of Unmanned Aerial Vehicles for Dynamic Site Layout Planning in Large-Scale Construction Projects"

_buildings, doi:10.3390/buildings11120602_

Round 1

Reviewer 1 Report

Dear Authors,

Your paper provides interesting insights associated with the use of UAV în civil construction.
Our minor suggestions are the following:
- please provide for the Materials and methods section the characteristics/attributes of the used drone and of the flight(s)
- our suggestion is to introduce a Discussion part in order to emphasize a debate starting from your results and associated with your topic
- please provide concrete limits of the applied methodology.

Author Response

Q1: Please provide for the Materials and methods section the characteristics/attributes of the used drone and of the flight(s)

Reply: We thank the reviewer for such a comment. We apologise for the confusion that we believe was caused by a failure to define the purpose of the article. The crucial point of the work is the development of a framework that integrates with the Site layout planning optimisation model to enable an enhanced construction site planning process using UAVs for site location mapping. We have now provided the details of the UAV used in Table 1, along with the flight mission.  We made changes in the Abstract and in Section 4 to reflect this.

Abstract: Construction sites are increasingly complex, and it is essential to implement mathematical optimisation techniques to increase the efficiency of their operations. In this context, the layout of the site influences productivity, safety, and efficiency. Dynamic site layout planning (DSLP) considers the adjustment of construction schedules on an evolving basis, allowing the relocation of temporary facilities according to the stages of the project. The main objective of this study is to develop a framework for integrating drones and their capacity for effective photogrammetry with a site layout planning optimisation model and Building Information Modelling (BIM) for automating site layout planning in large-scale projects, thus facilitating the application of the Dynamic Site Layout Planning concept. The mathematical model proposed is based on a Mixed Integer Programming (MIP) model which was employed to validate the framework on a realistic case study provided by an industry partner. Allocation constraints were formulated to ensure the placement of the facilities in feasible regions. Using information from the UAV, several parameters could be considered, including proximity to access-ways, distances between the facilities, and suitability of locations. Based on the proposed strategy, a layout was developed for each stage of the project, adapting the location of temporary facilities according to current trends. As a result, the use of space was optimised, and internal transport costs were progressively reduced.

Section 4:

An industry partner performed the photogrammetry part for the purposes of examining the integration of UAVs with site layout planning, using aDJI Inspire 1 v2.0, whose technical details are given in Table 1. As can be noticed from the technical details of the UAV, a simple drone was used which has a camera mounted that can capture the site images. The framework proposed thus does not require a sophisticated system and the basic requirements of a suitable UAV for the framework can be readily available anywhere in the world. An autonomous flight mission was defined using DJI GS Pro for the flight plan and the setting up of flight restrictions.

Table 1. UAV technical details

Parameter

Value

Dimensions

43.8 x 45.1 x 30.1 cm

Weight

2.93 kg

Camera resolution

UHD (4K): 4096 x 2160 p24/25

Speed

22 m/s (max)

Field of view

94º

Battery capacity

4500 mAh

Wind resistance

10 m/s

Q2: Our suggestion is to introduce a Discussion part in order to emphasize a debate starting from your results and associated with your topic

Reply: Thank you for your comment. We have now added a Discussion section.

  1. Discussion

During the execution of the case study, a number of points were observed. First, there were certain challenges related to the adoption of the method by the working team on the project. This was found to be due to the training needed in terms of drone operation, use of image processing to generate the required parameters, integration of point cloud with as-planned models for dynamic schedule validation and linking of data extracted from the BIM with the optimisation model. When it came to the drone operation, there were some misconceptions and misunderstandings in terms of operating duration of the drone. The research team made it clear that if the camera was set to capture an image every 2 seconds, then in the span of less than 10 mins, the entire construction site can be covered, and the resulting images that are stitched together to produce the point cloud would be of high resolution. The battery capacity of the UAVs adopted should therefore be at least 12 minutes roughly in duration, which is satisfied on most of the drones available on the market with similar specifications as Table 1. What was also noticed is that many of the engineers and architects working on the project had limited understanding of drone technology, along with their poor comprehension of the benefits of deploying operational research techniques to conduct an optimised dynamic site layout plan.

Another important challenge that is likely to be experienced on projects implementing the proposed framework is clutter; specifically, obstacles may obscure a clear vision of the camera, and this can block certain aspects of the site from being observed. The data processing step in terms of generating the point cloud model is high dependent on the amount of pictures that are captured and on the size of the project. For site layoyt planning problems, since the importance lies in the mapping of the outdoor environment, not much emphasis is placed on crating high resolution models for the building elements themselves, so long as outdoor site conditions can be mapped for pinpointing suitable locations available to position the facilities.

A key point to emphasise is that the method is very easy to implement and does not require the use of a sophisticated UAV types. Most drones available on the market have the minimum technical capacity required to enable such a framework to be implemented. It is also important to note the importance of the validation stage that is performed when the as-built point cloud model is contrasted with the as-planned BIM models; this step enables the verification of the progress of the project, thus allowing for a correct mapping of the stage of the project, and so an SLP problem can be solved that is explicit for that stage.

One final note to make is that since, the present work seeks an optimal solution to the proposed site layout planning problem through mathematical optimisation, a verified solution is reached once convergence in the solution algorithm is achieved (i.e. the gap between upper bound and lower bound on the solution is zero). It is impossible to cross-validate solutions on a large site as it would require tyring all combinations of possible facility-location allocations in reality, which no organisation would agree to due to the costly nature of the process.

Q3: Please provide concrete limits of the applied methodology.

Reply: Thank you for your suggestion. We have added the limitations of our study in the conclusion.

The limitations of the study are as follows: First, the framework developed, even though permits dynamic site layout planning to be performed, lacks in real-time capacity to capture site progress and make suggestions as to what needs to change in terms of the site layout for the next construction phase. Such capabilities can be permitted if machine learning algorithms are integrated with the UAV system, which was not within the scope of the existing study. Second, the proposed system is not fully automated in that it requires manual input for defining the flight mission, processing the images from the UAV to generate the point cloud, and identifying suitable location from the BIM model. To address such gaps requires coding all steps into an automated platform which is currently being undertaken by the authors.

Reviewer 2 Report

The authors propose an interesting framework for layout planning, which is an important and meaningful topic and has a good fit for the Buildings journal. However, I cannot be convinced by a method without cross-validation. In addition, the structure is a little bit confusing which makes the paper hard to follow, especially the Method section.

I suggested a major revision.

Author Response

Q1: I cannot be convinced by a method without cross-validation.

Reply: Thank you for pointing this out. We agree that cross-validation is a fundamental technique for evaluating predictive models. However, the present work seeks an optimal solution to the proposed problem through mathematical optimisation, which is proved once convergence in the solution algorithm is achieved. This is how all mathematical optimisation models in the literature verify their work. It is impossible to cross-validate on a large site as it would require tyring all combinations of possible facility-location allocations which no organisation would agree to due to the costly nature of the process. We clearly express this in the paper now.

Section 5:

It is also important to note the importance of the validation stage that is performed when the as-built point cloud model is contrasted with the as-planned BIM models; this step enables the verification of the progress of the project, thus allowing for a correct mapping of the stage of the project, and so an SLP problem can be solved that is explicit for that stage.

One final note to make is that since, the present work seeks an optimal solution to the proposed site layout planning problem through mathematical optimisation, a verified solution is reached once convergence in the solution algorithm is achieved (i.e. the gap between upper bound and lower bound on the solution is zero). It is impossible to cross-validate solutions on a large site as it would require tyring all combinations of possible facility-location allocations in reality, which no organisation would agree to due to the costly nature of the process.

Q2: The structure is a little bit confusing which makes the paper hard to follow, especially the Method section.

Reply: We thank the reviewer for such a comment. We have now re-configured the Method section to make it easier for the reader to follow. Specifically, the method section is now divided into 2 parts; the first part provides an overview of the proposed approach., while the second part elaborates on how locations can be delineated and how associated parameters can be calculated to embed in the site layout planning model.

Reviewer 3 Report

Comments and Suggestions for Authors

  1. Line 90: “The main objective of this study is to highlight the potential benefits and applications of integrating UAV and Building Information Modelling (BIM) for site layout planning in large-scale projects, through a framework employed to assimilate UAVs with Dynamic Site Layout Planning.” With this formulation of the objective, the reader should expect a comparison of the features revealed by the authors with articles those presented in Section 2. However, there is no such comparison.
  2. Line 157: “Typical site layout planning is performed based on 2D plans, involving the construction site’s dimensions.” One cannot agree with this statement. It was true 30 years ago. Currently, GIS databases and digital elevation models are used. GIS technologies deserve more detailed analysis in connection with the issues discussed here.
  3. The circuit shown in Figure 1 is fairly standard. It is necessary to indicate exactly what is the novelty of this scheme.
  4. The components shown in Figure 2 is also fairly standard. How photogrammetry is used. What are the features of the components?
  5. Section 4 should indicate how the proposed method uses the fact that the data is received from the drone.
  6. The conclusion does not explain how the main objective of this study is implemented: “The main objective of this study is to highlight the potential benefits and applications of integrating UAV and Building Information Modeling (BIM) for site layout planning in large-scale projects, through a framework employed to assimilate UAVs with Dynamic Site Layout Planning. "

Author Response

Q1: Line 90: “The main objective of this study is to highlight the potential benefits and applications of integrating UAV and Building Information Modelling (BIM) for site layout planning in large-scale projects, through a framework employed to assimilate UAVs with Dynamic Site Layout Planning.” With this formulation of the objective, the reader should expect a comparison of the features revealed by the authors with articles those presented in Section 2. However, there is no such comparison.

Reply: We thank the Reviewer for this insightful suggestion in improving the manuscript. In fact, the initially proposed objective was not aligned with the main point of the research. We now highlight the specific objective of the research

Abstract: Construction sites are increasingly complex, and it is essential to implement mathematical optimisation techniques to increase the efficiency of their operations. In this context, the layout of the site influences productivity, safety, and efficiency. Dynamic site layout planning (DSLP) considers the adjustment of construction schedules on an evolving basis, allowing the relocation of temporary facilities according to the stages of the project. The main objective of this study is to develop a framework for integrating drones and their capacity for effective photogrammetry with a site layout planning optimisation model and Building Information Modelling (BIM) for automating site layout planning in large-scale projects, thus facilitating the application of the Dynamic Site Layout Planning concept. The mathematical model proposed is based on a Mixed Integer Programming (MIP) model which was employed to validate the framework on a realistic case study provided by an industry partner. Allocation constraints were formulated to ensure the placement of the facilities in feasible regions. Using information from the UAV, several parameters could be considered, including proximity to access-ways, distances between the facilities, and suitability of locations. Based on the proposed strategy, a layout was developed for each stage of the project, adapting the location of temporary facilities according to current trends. As a result, the use of space was optimised, and internal transport costs were progressively reduced.

Section 1:

The AEC (Architecture, Engineering, and Construction) industry has been one of the most attractive and promising markets for UAVs [23; 43], and several researches have explored its potential in a wide range of applications, such as: building inspection [44], site mapping and surveying [27; 45], bridge inspection [46], progress monitoring, and site planning. However, despite the latter having been pointed out by Albeaino and Gheisari [47] as the second most recurrent application of UAVs in civil construction, few recent works are addressing the development of methodologies that support DSLP practically and systematically. The main objective of this study is to develop a framework for integrating drones and their capacity for effective photogrammetry with a site layout planning optimisation model and Building Information Modelling (BIM) for automating site layout planning in large-scale projects, thus facilitating the application of the DSLP concept. A case study based that showcases the implementation of Mixed Integer Programming (MIP) to solve the SLP was verify the applicability of the proposed approach.

Q2: Line 157: “Typical site layout planning is performed based on 2D plans, involving the construction site’s dimensions. ”One can not agree with this statement. It was true 30 years ago. Currently, GIS databases and digital elevation models are used. GIS technologies deserve more detailed analysis in connection with the issues discussed here.

Reply: We thank the Reviewer for this suggestion. We agree with this and have modified the paper to reflect this.

2.2. GIS and BIM for site layout planning

Typical site layout planning, for a long time, used to be performed based on 2D plans, involving simplified geometry. When the dynamics of the construction work are considered, a schedule is incorporated in the planning phase to generate dynamic site layouts across the various construction stages [63]. Thus, for generating dynamic site plans, it is essential to be able to generate up to date information for each stage of construction, so that changes that need to be implemented on the initial layout of the site can be applied [75]. Such information is often obtained manually by field engineers through on-land captured site images, daily reports, etc. Since the approach is based on ideal design information, the separate layouts generated for the construction stages can be different from layouts that are based on realistic representation of actual work being conducted. However, obtaining data for constantly assessing the deviations between as-planned and as-built progress, in order to generate more realistic site layout plans, seems like a tedious task for on-site engineers to carry out. It is the aim of the presented framework to provide the means for providing a more effective means of the process of obtaining the necessary information required to produce a dynamic layout particularly when it comes to delineating available locations. Figure 1 presents the phases undertaken to produce a dynamic layout for each stage of the construction process, via incorporation of on-site photos obtained from UAVs to generate appropriate facility locations.

Several technologies have been proposed to facilitate the planning and management of construction site layouts. Research using radio-frequency identification (RFID) and global positioning system (GPS) brought interesting contributions to this area of knowledge [64]. However, in recent years, the use of BIM models associated with GIS has shown constant interest amongst scholars in the field, especially in the construction sector [65-68]. According to [69], use of GIS allows the storage of location data referring to a region or facility that can be integrated with satellite images and digital elevation models (DEM), therefore enabling the analysis of the project’s evolution over time [70,71]. ZVia use of GIS, planners have a tool that helps them in the process of automating the planning of the site layout, modelling their spatial relationships and geometric conditions [72], and contributing considerably to increasing assertiveness in decision-making and reducing costs [65].

In addition, the use of BIM for site layout planning has been explored [76]. Incorporating BIM to solve the SLP problem has the advantage of simulating the physical model in a virtual environment [68], through parameterised information, hence allowing an accurate digital representation of the real object.

Lee and Lee [66] used BIM, GIS and Internet of Things (IoT) to develop a digital twin model for real-time logistics simulation in the construction industry. Pepe et al. [73] proposed the creation of a 3D GIS model of a cultural heritage site combining BIM, GIS and terrestrial laser scanner. Zhu and Wu [70] deepened operational knowledge between BIM and GIS files, developing a common geo-referencing approach for data integration. Liu et al. [67] explored the integration of 4D BIM and GIS during construction stage, resulting in the term GeoBIM. Khan et al. [68] explored the integration of BIM and GIS for modelling geotechnical properties and safe construction zones based on soil type, which was the same approach used by [72]. Irizarry et al. [74] integrated BIM and GIS to visually monitor supply chain in construction sites. Finally, [64] used GIS to develop a new methodology for risk assessment on construction sites.

Due to the inserts described above, the following references have been added:

  1. Abune’meh, M.; El Meouche, R.; Hijaze, I.; Mebarki, A.; Shahrour, I. Optimal construction site layou based on risk spatial variability. Automation in Construction 2016, 70, 167-177, doi:10.1016/j.autcon.2016.06.014
  2. Wei, J.; Chen, G.; Huang, J.; Xu, L.; Yang, Y.; Wang, J.; Sadick, A. BIM and GIS applications in bridge projects: A critical review. Applied Sciences 2021, 11, 6207, doi:10.3390/app11136207
  3. Lee, D.; Lee, S. Digital twin for supply chain coodination in modular construction. Applied Sciences 2021, 11, 5909, doi:10.3390/app11135909
  4. Liu, A.H.; Ellful, C.; Swiderska, M. Decision making in the 4th dimension – Exploring use cases and technical options for the integration of 4D BIM and GIS during construction. International Joural of Geo-Information 2021, 10, 203, doi:10.3390/ijgi10040203
  5. Khan, M.S.; Park, J.; Seo, J. Geotechnical property modeling and construction safety zoning based on GIS and BIM integration. Applied Sciences 2021, 11, 4004, doi:10.3390/app11094004
  6. Worboys, M.F.; Duckham, M. GIS: A computing perspective. CRC Press: Boca Raton, FL. USA, 2004.
  7. Zhu, J.; Wu, P. A common approach to geo-referencing building models in industry foundation classes for BIM/GIS integration. International Joural of Geo-Information 2021, 10, 362, doi:10.3390/ijgi10060362
  8. Liu, X.; Shannon, J.; Voun, H.; Truijens, M.; Chi, H.L.; Wang, X. Spatial and temporal analysis on the distribution of active radio-frequency identification (RFID) tracking accuracy with the kriging method. Sensors 2014, 14, 20451-20467, doi:10.3390/s141120451
  9. Sarasanty, D. Safety hazards identification of construction site layout based on geographic information system (GIS). International Journal on Advanced Science Engineering Informetion Technology 2020, 10(5), 2021-2027.
  10. Pepe, M.; Constantino, D.; Alfio, V.S.; Restuccia, A.G.; Papalino, N.M. Scan to BIM for the digital management and representation in 3D GIS environment of cultural heritage site. Journal of Cultural Heritage 2021, 50, 115-125, doi:10.1016/j.culher.2021.05.006
  11. Irizarry, J.; Karan, E.P.; Jalaei, F. Integrating BIM and GIS to improve the visual monitoring of construction supply chin management. Automation in Construction 2013, 31, 241-254, doi:10.1016/j.autcon.2012.12.005
  12. Cheng, M.; Chang, N. Dynamic construction material layout planning optimization model by integrating 4D BIM. Engineering and Computers 2019, 35(2), 703-720, doi:10.1007/s00366-018-0628-0
  13. Le, P.L.; Dao, T.M.; Chaabane, A. BIM-based framework for temporary facility layout planning in construction site: a hybrid approach. Construction Innovation 2019, 19(3), 424-464, doi:10.1108/CI-06-2018-0052
  14. Nguyen, P.T. Construction site layout planning and safety management using fuzzy-based bee colony optimization model. Neural Computing and Applications 2021, 33, 5821-5842, doi:10.1007/s00521-020-05361-0
  15. Han, K.K.; Golparvar-Fard, M. Automated monitoring of operation-level construction progress using 4D bim and daily site photologs. Proceedings of the Construction Research Congress: Construction in a Global Network (CRC 2014), 2014, Atlanta, EUA, 2014.
  16. Bang, S.; Kim, H.; Kim, H. UAV-based automatic generation of high-resolution panorama at a construction site with focus on preprocessing for image stitching. Automation in Construction 2017, 84, 70-80, doi:10.1016/j.autcon.2017.08.031
  17. Anwar, N.; Najam, F.A.; Izhar, M.A. Construction monitoring and reporting using drones and unmanned aerial vehicles (UAVs). Proceedings of the Tenth International Conference on Construction in the 21st Century (CITC-10), 2018, Colombo, Srilanka, 2018.

Q3: The circuit shown in Figure 1 is fairly standard. It is necessary to indicate exactly what is the novelty of this scheme.

Reply: Thank you for pointing this out. We have specifically highlighted the novelty of the work, with references that support our argument.

Section 2:

The novelty of the proposed framework in this study for the dynamic SLP can be highlighted as follows. Firstly, the proposed approach combines site layout planning with UAV photogrammetry to identify locations on a construction site for positioning of temporary facilities; in SLP studies, it is common to assume that the construction site is established as a 2D rectangular space discretised into a grid of candidate locations [53]. It is also common to assume that the locations are a priori declared [77]. Secondly, the proposed framework contributes to the automation of the dynamic SLP in an objective way without emphasis on subjective location decisions, for the dynamic SLP. Specifically, this is the first attempt to link images captured of the site from UAV with the mathematical optimisation for the purpose of available location identification, through site reconstruction via point clouds. Thirdly, BIM is linked to the images captured via superposition of the as-planned model with the as-built images using well-established techniques [78], thus aiding in the tracking of progress and ensuring that locations mapped as available are up to date. Fourth, the proposed framework allows for the automatic calculation of the several optimisation model parameters associated with the locations identified for facility positioning, including the distances between locations, and the costs of having locations available for the temporary facilities.

                In addition, an important contribution of our study is the systematic procedure established in Figure 3, to showcase the process of identifying the locations for locating the temporary facilities, and generating parameters associated with the locations in the SLP model. We now have a specific section that elaborates further on Figure 3 in Section 3.2.

3.2 Generation of Locations for SLP

A key contribution in this study is describing how UAVs can be implemented for the purpose of generating the appropriate locations available for placement of temporary facilities on a construction site, along with the computation of location parameters in the mathematical optimisation model involved. The steps involved in the generation of the dynamic site layout of a large construction site are summarised in the flowchart of Figure 3. The process addresses integrating UAV with the as-planned models such as BIM so that site layout mapping and updating across the different stages of a project is achieved.

Incorporating UAV technology for site layout planning enables the consideration of physical measurements that are directly extracted from images taken at a set interval by a mounted camera. Other aspects accounted for include the consideration of varying site conditions in terms of location availability that can quickly be incorporated in the site layouts of each stage of the project, thus leading to a dynamic generation of the set of available locations for facility positioning throughout the construction project phases. The approach proposed relies on use of UAV as they are regarded as an efficient and cost-effective alternative technology for monitoring the evolving process of construction projects. Assessing the impacts of ongoing work on the location of temporary facilities can be fully examined when regular updates of the occupied areas on the site are available. Being able to closely monitor changes on the construction site due to applications of the UAV system also offers the ability to update and verify the frequency of travel between facilities on the construction site. This, as explained later in the paper, comprises a vital part of the optimisation model used for SLP.

Figure 2. Components of the UAV system

As an initial step in Figure 3, it is vital to supply the UAV system with a waypoints digital XML file for delineating the flight trajectory path. The path should be programmed such that essential coordinate points in the physical space, including longitude, latitude, and altitude coordinates, are incorporated in the UAV’s travel trajectory. A preliminary set of coordinate points can be obtained from initial 2D plans of construction works at the early stages of construction, and from the as-planned 3D BIM of the project to be constructed. Waypoints are defined based on important field views that form the border of the area to be investigated by the UAV. Once acquired, the waypoints may be subject to updates based on a comparative study conducted to assess deviations in the planned vs actual progress of works.

Embedded within the UAV system is a navigation routine that is comprised of a global positioning system (GPS), making use of GNSS and an inertial navigation system (INS). Such systems enable the autonomous tracking of the defined waypoints. Apart from its use in navigation, incorporating a GPS receiver within the UAV system also serves the purpose of geo-spatially referencing data acquired from captured images. This is imperative for the construction of ortho-photos from which direct measurements can be made for the optimisation model’s parameters, such as distances, areas, and coordinated of the centroids of the locations used to solve the SLP; this step is important too for exporting of data to digital elevation models (DEM) that enable an estimate of the cost parameter in the optimisation model required to prepare a location for hosting a temporary facility. Referencing the data acquired from images captured therefore is necessary as it enables the identification of desirable locations for temporary facilities and the production of an updated site layout configuration that is labelled based on accurate coordinates.

Processing of images captured by the mounted camera is done to produce four types of data visualizations that will be imperative for the generation of several location parameters embedded in the SLP model. For generating 3D point clouds, Scale Invariant Feature Transform (SIFT) is applied to allow for the detection of key feature points within the delineated zone on the construction site. These point clouds are directly geo-referenced data points, generated from densely grouped coordinates. From the point clouds, the DEM and the ortho-photos can be formed. Utilising the ortho-photos and the DEM, locations on the construction site can be analysed based on distance measures between the locations and material demand points, and suitability of terrain for construction of the temporary facility can then be used to generate the cost associated with each location on site. The costs generated relates to the terrain of the location in terms of the setup costs required to be expended for placement of a temporary facility there. This cost is later on embedded in the objective function as a parameter associated with each location identified, thus allowing for decision-makers to minimise the total monetary cost of setting up the site layout.

Figure 3. Flow chart depicting site layout updating process

For site layout, it is important to identify any safety issues and hazards linked with the construction works. These can be pinpointed from the processed images obtained from the UAV technology. Special safety requirements essential to some temporary facility types, for example, the placement of engineers’ offices as far apart from falling hazards as is permitted, can be accounted for, through viewing suitability of locations and their closeness to hazardous areas on ortho-photos. A negative weighting can then be assigned to these locations in the optimisation model parameters generated, to avoid locating facilities in the hazard-deemed regions.

To infer how the progress on site compares with the planned schedule the 3D point clouds processed from aerial images taken by the UAV system can be superimposed on the building information model. Each building element of a structure undergoing construction can be back-projected onto the processed images for occupancy-based and material-based appearance modelling. Detection of any mismatch between as-built data, acquired from UAV images, and as-planned models, such as BIM, will entail the updating of the planning schedule and as-planned models to match actual site progress. This will then determine whether the construction site layout should be updated, or whether certain facilities need to remain in particular regions for a given period. Occlusions present in the captured images are dealt with when back projecting BIM elements against the images, and this can lead to the specific information being modelled and fed into the optimization module for determining whether the progress of a certain element/ group of elements requires the repositioning of facilities. Gaps in the information that can be gleaned from both spectra can therefore be filled when contrasts are made between as-built and as-planned designs.

Q4: The components shown in Figure 2 is also fairly standard. How photogrammetry is used. What are the features of the components?

Reply: Thank you for this observation. Our purpose for incorporating Figure 2 was to demonstrate how generic information generated from UAVS can be easily integrated into the SLP, in order to contextualise to the reader the possibilities of enhancing SLP models through the data that is collected from UAVs. In this sense, unlike works that focus specifically on practical applications of UAVS, our study focused on the development of a framework that demonstrates how data can be easily integrated from UAVs into the SLP model, for better representation of the parameters of the model associated with the location decision variable. We now explain this in the manuscript.

Section 3.1:

Figure 2 therefore demonstrates how generic information that is generated from UAVs can be easily integrated into the SLP, to enhance the accuracy of the location parameters embedded in the SLP models. In this sense, unlike works that focus specifically on practical applications of UAVS, our study focused on the development of a framework that demonstrates how data can be easily integrated from UAVs into the SLP model, for better representation of the parameters of the model associated with the location decision variable.

Q5: Section 4 should indicate how the proposed method uses the fact that the data is received from the drone.

Reply: Thanks to the Reviewer for this suggestion. Our manuscript has been updated to reflect this. Please see Sections 3, 4 and 5.

Q6: The conclusion does not explain how the main objective of this study is implemented: “The main objective of this study is to highlight the potential benefits and applications of integrating UAV and Building Information Modeling (BIM) for site layout planning in large-scale projects, through a framework employed to assimilate UAVs with Dynamic Site Layout Planning. "

Reply: Thank you for pointing this out. We have now updated our conclusion.

  1. Conclusion

A framework was presented for incorporating a UAV system to aid in the process of site layout mapping and updating in large construction sites. The UAV system captures overlaid images from different positions within the site, as determined from an input waypoints file. These images are processed to produce 3D point clouds, ortho-photos, digital surface models, and consequently 3D models of the constructed structures. Data on the locations available for positioning facilities, including their centroids, associated costs and distance parameters extracted from the 3D point clouds, ortho-photos and digital surface models, were then embedded in a MIP optimisation model formulated to minimise the total material handling costs and the temporary facility construction setup costs. A single layout is produced for each stage of the construction process, hence rendering the SLP problem a dynamic one. The use of UAVs in the Site layout planning problem addressed allows the generation of accurate locations available across the stages of the project, via effective coverage of large-scale sites efficiently and economically. This allows the BIM model to be updated with greater frequency, consistency, and accuracy, reducing the chances of human error in the mapping processes. Thus, instead of static site updates of the site at pre-determined periods of spaced time, dynamic site layouts are carried out, so that the model and the relocation need of the temporary facilities can be evaluated and the decisions made practical in real time. Applications of the framework were illustrated on a large civil works project.

The limitations of the study are as follows: First, the framework developed, even though permits dynamic site layout planning to be performed, lacks in real-time capacity to capture site progress and make suggestions as to what needs to change in terms of the site layout for the next construction phase. Such capabilities can be permitted if machine learning algorithms are integrated with the UAV system, which was not within the scope of the existing study. Second, the proposed system is not fully automated in that it requires manual input for defining the flight mission, processing the images from the UAV to generate the point cloud, and identifying suitable location from the BIM model. To address such gaps requires coding all steps into an automated platform which is currently being undertaken by the authors.

Round 2

Reviewer 2 Report

I appreciate the authors' effort to improve the paper.

Reviewer 3 Report

You have improved manuscript